# A Community Study of SARS-CoV-2 Detection by RT-PCR in Saliva: A Reliable and Effective Method

**DOI:** 10.3390/v14020313

**Published:** 2022-02-02

**Authors:** Filippo Fronza, Nelli Groff, Angela Martinelli, Beatrice Zita Passerini, Nicolò Rensi, Irene Cortelletti, Nicolò Vivori, Valentina Adami, Anna Helander, Simone Bridi, Michael Pancher, Valentina Greco, Sonia Iolanda Garritano, Elena Piffer, Lara Stefani, Veronica De Sanctis, Roberto Bertorelli, Serena Pancheri, Lucia Collini, Erik Dassi, Alessandro Quattrone, Maria Rosaria Capobianchi, Giancarlo Icardi, Guido Poli, Patrizio Caciagli, Antonio Ferro, Massimo Pizzato

**Affiliations:** 1Department of Cellular, Computational and Integrative Biology, University of Trento, 38123 Trento, Italy; filippo.fronza.o@gmail.com (F.F.); nelli.groff@unitn.it (N.G.); angela.martinelli@unitn.it (A.M.); beatrice.passerini@unitn.it (B.Z.P.); nicolo.rensi@unitn.it (N.R.); irene.cortelletti@gmail.com (I.C.); nicolo.vivori@alumni.unitn.it (N.V.); valentina.adami@unitn.it (V.A.); anna.helander@unitn.it (A.H.); simone.bridi@unitn.it (S.B.); michael.pancher@unitn.it (M.P.); valentina.greco@unitn.it (V.G.); sonia.garritano@unitn.it (S.I.G.); elena.piffer@unitn.it (E.P.); lara.stefani@unitn.it (L.S.); veronica.desanctis@unitn.it (V.D.S.); roberto.bertorelli@unitn.it (R.B.); erik.dassi@unitn.it (E.D.); alessandro.quattrone@unitn.it (A.Q.); etruscus50@gmail.com (P.C.); 2Azienda Provinciale per i Servizi Sanitari, 38123 Trento, Italy; serena.pancheri@apss.tn.it (S.P.); lucia.collini@apss.tn.it (L.C.); antonio.ferro@apss.tn.it (A.F.); 3National Institute for Infectious Diseases “L. Spallanzani”, 00149 Roma, Italy; maria.capobianchi@inmi.it; 4Department of Health Sciences, University of Genova, 16132 Genova, Italy; icardi@unige.it; 5Vita-Salute San Raffaele University, 20132 Milano, Italy; poli.guido@hsr.it

**Keywords:** COVID-19, saliva testing, molecular diagnosis, SARS-CoV-2 detection

## Abstract

Efficient, wide-scale testing for SARS-CoV-2 is crucial for monitoring the incidence of the infection in the community. The gold standard for COVID-19 diagnosis is the molecular analysis of epithelial secretions from the upper respiratory system captured by nasopharyngeal (NP) or oropharyngeal swabs. Given the ease of collection, saliva has been proposed as a possible substitute to support testing at the population level. Here, we used a novel saliva collection device designed to favour the safe and correct acquisition of the sample, as well as the processivity of the downstream molecular analysis. We tested 1003 nasopharyngeal swabs and paired saliva samples self-collected by individuals recruited at a public drive-through testing facility. An overall moderate concordance (68%) between the two tests was found, with evidence that neither system can diagnose the infection in 100% of the cases. While the two methods performed equally well in symptomatic individuals, their discordance was mainly restricted to samples from convalescent subjects. The saliva test was at least as effective as NP swabs in asymptomatic individuals recruited for contact tracing. Our study describes a testing strategy of self-collected saliva samples, which is reliable for wide-scale COVID-19 screening in the community and is particularly effective for contact tracing.

## 1. Introduction

SARS-CoV-2, the coronavirus causing COVID-19, has spread in all continents since the end of 2019, quickly becoming pandemic. The infection causes a respiratory syndrome which can be lethal, especially in individuals affected by other pathologies and in the elderly population. The widespread diffusion of the virus exposed the healthcare systems worldwide to an unprecedented pressure, quickly overpowering hospitals and healthcare professionals. In order to control the spread of the virus, to protect the individuals most at risk and to preserve the functionality of the healthcare system, the capability to promptly track the presence of SARS-CoV-2 in the community remains critical [1].

For a long time, the gold standard to reliably diagnose respiratory infections has been the analysis of secretions from the mucosa of the upper respiratory epithelium collected by nasopharyngeal or oropharyngeal swabs. For the diagnosis of SARS-CoV-2, maximum sensitivity is ensured by detecting the viral RNA in the swab using RT-PCR [2]. However, the success of the diagnosis relies on the correct procedure for collecting the specimen, which, in the case of nasopharyngeal samples, requires the introduction of the swab through the nasal septum to reach the epithelium of the nasopharynx, causing discomfort to the individuals being tested [3]. While the correct procedure is normally performed by skilled healthcare professionals, some countries (e.g., the UK) implemented a policy of swab self-administration to favour testing scale-up. Such practice is debated as other countries, such as Italy, currently consider swab self-collection too unreliable and, therefore, employ thousands of trained professionals for community testing. In such cases, swab collection also poses a biohazard risk for the personnel collecting the specimens who are directly exposed to the aerosol generated by the individual being tested. To overcome these problems, the possibility of using saliva for COVID-19 testing was explored soon after the virus was declared pandemic as an alternative to nasopharyngeal swabs [4].

Several studies investigated the presence of SARS-CoV-2 in oral fluids at different stages of the infection by RT-PCR, comparing the sensitivity of virus detection in saliva and in pharyngeal swabs. The current literature provides a consensus suggesting that viral load found in saliva is adequate for a reliable diagnosis, with several investigations reporting more efficient detection of SARS-CoV-2 in saliva than nasopharyngeal swabs [5,6,7]. Despite the encouraging evidence, adoption of wide-scale saliva testing remains hampered by the lack of standardisation of the method of collection and the analytical procedure [8]. Available devices for collecting saliva can be very different, potentially impacting the sensitivity of detection. Popular methods include the absorption of saliva on a pad from which the fluid has to be mechanically eluted or drooling into a test tube [9]. In either case, the degree of viscosity, which varies among subjects, can hinder downstream processing, interfering with pipetting, causing contamination and impacting on the efficiency of the RNA extraction.

Here, we describe the result of a large diagnostic effort implemented within an academic setting (the University of Trento, Trento, Italy) to assist the local health service with testing and screening activities for SARS-CoV-2 infection in the general population. To this end, a workflow for the mass-scale analysis of saliva samples was established, starting with the design of a system that facilitates the autonomous and safe collection of saliva and the semi-automation of the downstream processing of samples. This strategy was tested in the context of the general population, including symptomatic and asymptomatic individuals who were subjected to both saliva and nasopharyngeal swab analysis, allowing for a direct comparison.

## 2. Materials and Methods

### 2.1. Study Population

Subjects were recruited at a drive-through testing facility between 25 March and 23 April 2021. The cohort included individuals from the general population undergoing COVID-19 testing for different reasons. Some individuals were tested because they presented with respiratory symptoms, others had already been diagnosed with COVID-19 and they were tested to confirm their recovery, others were tested for contact tracing and, finally, a group of healthcare operators was undergoing regular, routine screening (Table 1). The study received prior approval by the Ethical Committee of the local health authority (Azienda Provinciale per i Servizi Sanitari of the autonomous province of Trento). Informed consent was obtained from all participants undergoing testing.

### 2.2. Sample Collection

All subjects enrolled underwent testing of NP swabs and saliva. While waiting in line for collection of the NP swab by a healthcare professional, a saliva collection kit (Comedical, Trento, Italy) was distributed to the participants. Each individual collected the saliva autonomously after reading or watching the instructions provided with the kit (https://www.covidsaliva.it/ accessed 25 March 2021). Inclusion in the study required participants to not eat, drink or smoke for at least 30 min prior to saliva collection. The NP swabs were acquired immediately after saliva samples were collected. More than one healthcare professional acquired the NP samples from different subjects. Both saliva and NP swabs were transported to the laboratory the same day, stored at 4 °C overnight and processed for analysis the following day.

### 2.3. The Analytical Process

All liquid handling was performed with robotised, multichannel pipettes with adjustable tip spacing (Assist Plus, Integra Biosciences, Zizers, Switzerland). Saliva samples were first fluidified with the addition of 1 mL DTT 6.5 mM in water directly into the collection tubes followed by pipetting up and down 10 times for resuspending the reducing solution. After this operation, the saliva was sufficiently fluidified for further processing. Saliva and NP samples were thereafter transferred to 96-well DeepWell™ plates (Thermofisher, Scientific, Waltham, MA, USA) and extraction performed using the MagMAX^TM^ Viral/Pathogen Nucleic Acid Isolation Kit (Thermofisher, Scientific, Waltham, MA, USA) following the manufacturer’s instruction. After addition of the extraction mix, proteinase K was also added, as indicated by the manufacturer’s instructions, a step that was found to be crucial for the efficiency of the RNA extraction from the saliva. Extraction was operated with a KingFisher™ Flex Purification System (ThermoFisher, Scientific, Waltham, MA, USA) and automatic extractor, and RNA was eluted in 50 µL of RNAse-free water. PCR was set up on 96-well plates using a dedicated, robotised, multichannel pipette with adjustable tip spacing (Assist Plus, Integra, Biosciences, Switzerland). RT-PCR amplification was performed on CFX96 thermocyclers (BioRad, Hercules, CA, USA) using the Novel Coronavirus Real Time Multiplex RT-PCR Kit (Liferiver Bio-Tech, La Jolla, CA, USA) following the manufacturer’s instructions. During the period of our investigation, the largely predominant SARS-CoV-2 variant in Italy was alpha (B1.1.1.7), efficiently detected by the RT-PCR kit adopted for the study. Amplification results were analysed using CFX Maestro software with the single threshold method and Ct calculated by placing the threshold just above the signal given by the molecular grade water negative control sample. Diagnosis was based on the detection of RdRP gene within a threshold value of 40. The kit provides an internal extraction control to be added to each sample, which was used to monitor the efficiency of extraction. A custom-made middleware (Olos, Stardata, Ascoli Piceno Italy) was implemented to track the samples through the process, to guide the operators through the analytical phases, to retrieve the amplification results and to communicate the data to the healthcare service repository.

### 2.4. Statistical Analysis

The quantification of agreement (concordance) by kappa, the Pearson correlation coefficient, the confidence intervals and the standard deviations were calculated using Prism (GraphPad Software, San Diego, CA, USA).

## 3. Results

### 3.1. Design of an Analytical Workflow Allowing Large-Scale Screening of SARS-CoV-2 in Saliva

With the scope of assisting the safe self-collection of saliva by the users and, at the same time, facilitating high processivity of the downstream molecular analysis, a novel saliva collection device was first designed. Standard 13 mm × 80 mm tubes already widely used for collecting nasopharyngeal swabs were equipped with an insert functioning as a straw for the deposition of saliva (Figure 1a). To prevent contamination of the exterior of the tube with oral fluid and avoid the potential dispersion of contaminated material in the environment, the insert was made to fully enter and remain locked inside the tube after saliva collection. The straw was designed to be compatible with the direct introduction of 1250 µL pipette tips for withdrawing the oral fluid, avoiding the need for further processing steps, such as centrifugation, before proceeding directly with the analytical protocol.

Subjects enrolled in the study were instructed to collect between 0.5 and 1 mL of saliva into the tube, as indicated by labels on the tube itself (Figure 1a). The tubes with saliva were mounted on a rack and processed using a robot-operated, multichannel pipette with adjustable tip spacing, allowing the deposition of the oral fluid into a standard 96-well formatted plate (Figure 1b). To address the potentially high viscosity of saliva, 1 mL of a 6.5 mM DTT solution was first added to the samples using the same robotised pipette. This addition proved very effective at increasing fluidity, facilitating automatic pipetting and preventing the formation of threads of saliva on pipette tips, which could possibly cause cross-contamination. Based on the efficiency of the extraction of an internal control added to the samples being processed, DTT treatment was found compatible with the effective recovery of nucleic acids in every sample analysed. Saliva samples were automatically formatted into multi-well plates for downstream RNA extraction and RT-PCR, operated in batches of 96 (Figure 1b). The sample check-in required to enter the analytical line, the tracking of the samples through the different steps of the process, the evaluation of results and the final reporting to the health service repository was managed by a custom-made middleware which was instrumental for coordinating and instructing the operators through the different analytical phases. The analytical cycle, from the addition of DTT to the samples to the completion of the RT-PCR, lasted 3 h, with a processivity of 384 saliva samples per hour per production line.

### 3.2. Global Comparison of SARS-CoV-2 Detection in Saliva and NP Swabs

NP swabs and salivary samples were collected from 1025 individuals coming to a public drive-through testing facility in Trento, Italy. NP swabs were administered by healthcare workers while saliva was collected autonomously by each person enrolled in the study. Among the people investigated were symptomatic and asymptomatic subjects from the general population, as well as healthcare professionals undergoing routine periodic screening (Table 1). Saliva samples and NP swabs were analysed using the same RNA extraction method and RT-PCR kit and compared based on the detection of the RdRP gene over 40 PCR cycles. The amount of saliva collected by each subject was the only criterion adopted for excluding samples from analyses. Tubes containing visibly less than 0.5 mL of oral fluid were removed from further processing. Out of 1025 samples collected, only 22 (2.1%) were discarded based on this criterion.

Out of 1003 individuals tested, the number of SARS-CoV-2 positive saliva samples was 312 (31%) while SARS-CoV-2 was detected in 344 (34.3%) NP swabs (Figure 2a). However, the total number of positive subjects detected by either the saliva or the nasopharyngeal test methods was 419 (41.8%), with 237 samples found positive and 584 samples negative with both tests, giving a total concordance of 81.8% with a kappa index of 0.588 (Figure 2a).

Considering the NP swabs as reference, the global sensitivity of the saliva test was 68.9% (237/344) with a specificity of 88.6% (584/659). By assessing all subjects in which SARS-CoV-2 was detected, either in the saliva or in the NP swabs, the sensitivity of detection was 74.5% (312/419) with the saliva and 82.1% (344/419) with the NP swabs; therefore, moderately higher sensitivity was indicated for the latter.

When the Ct values of the positive samples were compared (Figure 2b), the mean values obtained with the two methods were similar (31.60 for NP and 32.19 for saliva). Overall, a strong and statistically significant correlation was found between Cts obtained with the two techniques (*r* = 0.6994 with *p*-value < 0.0001), indicating a general trend with individuals with high viral loads in the NP swabs presenting also with a high viral load in saliva (Figure 2c). However, in 182 subjects, SARS-CoV-2 was discordantly detected in the two types of samples; 75 individuals tested positive in the saliva and negative in the NP swabs (NP−/S+), while 107 subjects tested positive in the NP swab and negative in the saliva (NP+/S−, Figure 2a). The overall comparison of the Ct values from these discordant samples revealed similar mean Ct values. However, while all NP+/S− samples produced values higher than 29, a cluster of seven NP−/S+ samples was clearly different from the remaining samples, characterised by exceptionally low Ct values ranging between 21 and 26 (dotted rectangle in Figure 2d), irrespective of the group of individuals being tested. Therefore, some individuals with an exceptionally high viral load in the saliva failed to test positive in the NP swab (Figure 2d).

### 3.3. Saliva and NP Swabs Give Concordant Results in Symptomatic Subjects and in Individuals Recruited from Contact Tracing

The subjects enrolled in the study included individuals undergoing testing for having been recently in contact with infected persons (contact tracing), convalescent individuals testing to confirm their recovery, healthcare professionals for routine screening or symptomatic individuals suspected of infection (Table 1). Positive individuals recruited for contact tracing are likely to represent the initial stages of the infection, in contrast to convalescent subjects who are tested after clinical resolution of the infection.

We compared the performance of saliva and NP swabs across all groups (Figure 3). Our study included a total of 32 individuals that presented with respiratory symptoms. Remarkably, though the sample was numerically small, both assays identified the same 15 positive individuals (making 46.9% of the total) with a concordance of 100% (Figure 3a). In contrast, among the 971 asymptomatic subjects belonging to the different groups, a higher number of infections was detected in NP swabs (327) than in saliva (295), with 81.26% concordance between the two methods (Figure 3b). Results were further compared in the different groups of asymptomatic subjects. Among 442 convalescents, NP testing revealed 276 positive cases, 38 more than those detected in saliva (235), resulting in 64.93% concordance with a weak kappa index (0.287). In contrast, among 193 subjects recruited from contact tracing, more positive cases were revealed in saliva than NP swabs (65 vs. 56), with a strong concordance of 89.12% and a kappa index of 0.748. This result suggests that, while the oral fluid provides a less sensitive biological specimen than NP swabs for detecting viral RNA during convalescence, saliva performs at least as well as NP swabs during the early stages of the infection.

When Cts were compared in each group, values in saliva were generally 1–2 PCR cycles higher than in NP swabs (Figure 3e). However, Ct values from most contact tracing NP and saliva samples were 5–6 cycles lower than those observed in most convalescent subjects (Figure 3e), indicating that both saliva and NP swabs contain a much higher viral load in most individuals during the initial phases after they contract the infection. Subjects where viral RNA was detected only in one type of specimen (NP+/S− and NP−/S+) produced, on average, high Ct values, ranging between 32–35 PCR cycles, indicating that discordance between NP swabs and saliva is mainly restricted to low viral loads.

## 4. Discussion

To control the spread of SARS-CoV-2, the implementation of an efficient strategy to track the virus in the population is crucial [1]. To this end, saliva represents an ideal tool, given that it is easily accessible and can be self-collected with non-invasive procedures without involving specialised personnel. The method used to collect saliva from individuals to be tested could crucially affect the efficiency of detection and the processive capacity of the downstream analyses [9]. For example, some devices adopting absorbing pads to collect saliva require an elution step which could affect viral RNA extraction and add a labour-intensive step to the process. Here, we adopted a novel device which facilitates the self-collection of saliva, as well as high processivity in the laboratory. Excess viscosity is another a major issue commonly encountered when processing saliva since it interferes with the correct pipetting of the oral fluid. We show that the simple addition of a reducing agent (DTT) can effectively reduce viscosity while being compatible with the high-throughput downstream process. Semi-automation was implemented to address the need of high-throughput analysis, allowing reduced human intervention while keeping the analytical line flexible and compatible with reagents from different providers. Of note, the instrumentation adopted represents a moderate financial investment compared with fully automated systems currently available on the market and can be easily converted to different applications when SARS-CoV-2, large-scale testing is no longer required.

In addition to testing a novel collection device and a streamlined analytical procedure, our research, compared with most other studies (for a review, see [6]), involved a larger and more diverse cohort from the general community, with subjects who underwent testing for different reasons. The strategy described here was applied to a realistic context of community screening and confirmed that it can facilitate scale-up analysis by favouring safe self-collection of the oral fluid. Of note, viral RNA in saliva is stable for several days at room temperature after collection [10,11] without the use of transport media. We, therefore, envisage a situation where the collection devices are distributed to the individuals requiring screening who can collect the sample at home and return it safely to the testing laboratory. While not included in this study, our experience confirms that the collection device used here also facilitates the testing of children and makes the mass screening at schools easier and reliable.

The parallel examination of saliva and nasopharyngeal specimens from 1003 individuals revealed discordant detection of SARS-CoV-2 in 182 subjects, a result consistent with most data in literature, as reported by a review that analysed 58 studies published since the start of the pandemic [6]. Such a level of discordance between the two tests indicates that no approach can diagnose the infection in 100% of cases. However, stratification with different groups of individuals recruited in the study revealed interesting differences in the efficiency of viral detection by the two tests. SARS-CoV-2 was identified with 100% concordance in saliva and NP swabs among symptomatic patients. A good concordance was also observed among asymptomatic individuals recruited for contact tracing, with a slightly higher number of positive samples detected in saliva than in NP swabs. In contrast, poorer concordance was observed in convalescent subjects with a higher number of SARS-CoV-2 positive samples detected in NP swabs than in saliva. Our result, therefore, suggests that the oral fluid is an efficient matrix for the detection of the virus during the early stages of the infection and in the presence of symptoms, rather than late during convalescence. These outcomes agree with other studies that have ascertained that, soon after infection, SARS-CoV-2 can be abundantly detected in saliva [4,12,13], in line with evidence that indicates that the epithelium associated with the salivary glands is an early site of virus replication [14,15,16]. These results also agree with another recent investigation which confirmed that SARS-CoV-2 is more readily detected in saliva [17].

The poorer diagnosis of SARS-CoV-2 in saliva of convalescent individuals parallels the lower viral load observed in this group, with Ct values 5–6 cycles higher than those observed in contact tracing samples (see Figure 3e). One possibility is that the discordance between NP swabs and saliva in convalescents could reflect the lower capacity of the analytical process to reveal the presence of the viral RNA in the oral fluid, which would have a bigger impact in samples with viral loads closer to the limit of PCR detection. Alternatively, the discordance could be explained by a selectively lower amount of viral RNA in saliva during the convalescent phase compared to NP samples. Interestingly, it has been observed that the majority of NP swabs derived from clinically resolved infections, though positive, do not harbour virions that can be isolated in culture [18], but rather contain residual non-replicative RNA [19]. The lower detection in the saliva of convalescent subjects could, therefore, reflect the lower level of active viral replication during the late phase of the infection. In this case, the oral fluid could be a more selective indicator of the presence of infectious virus, thus, preventing convalescent subjects from undergoing unnecessary, long quarantine periods. Viral isolation studies from saliva and NP swabs are, therefore, warranted to test this hypothesis.

Interestingly, among individuals that tested positive only in saliva, a cluster was characterised by a particularly high viral load (Ct ranging between 21 and 26). Failure to detect SARS-CoV-2 in the NP swab of individuals with such high viral load in saliva was not associated with a specific subgroup of subjects and likely mirrors the quality of the NP samples collected by the healthcare operator. This is an indication that the saliva sample is less affected by the variability associated with the collection procedure and is, therefore, a more reliable diagnostic sample than the NP swab. This aspect could have greater implications if the NP swabs are self-administered by inexperienced users.

In conclusion, given that the droplets and aerosol created by saliva are the main viral transmission route [20], it is reasonable to postulate that individuals who shed a high amount of virus in the saliva are also the most contagious and the most crucial to identify in the community. By showing the efficacy and reliability of saliva testing in asymptomatic individuals and given that saliva can be safely self-collected at home, our study supports the use of the oral fluid for widespread COVID-19 testing in the community. Incidentally, while our study was performed when the SARS-CoV-2 alpha variant was predominant, the omicron variant, highly transmissible and with a preferential tropism for the upper respiratory tract [21,22,23,24], became globally widespread at the time this article was written. Since omicron is more efficiently detected in saliva than in nasal swabs compared to other variants [25,26], the use of the oral fluid for diagnosing active SARS-CoV-2 infections in the population is further warranted.

## Figures and Tables

**Figure 1 viruses-14-00313-f001:**
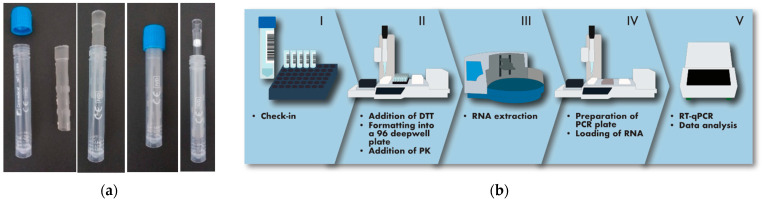
Description of the organisation of the analytical process. (**a**) The saliva collection device used in this study consists of a standard 13 mm tube provided with an insert to aid the deposition of saliva, which remains inside the tube without interfering with the introduction of pipette tips. (**b**) The analytical process begins with the tubes being prepared on racks (I). They are accepted by a robotised, multichannel pipettor with adjustable tip spacing (II), which produces 96-well formatted plates for batched RNA extraction (III). PCR plates are then assembled with a dedicated, robotised pipettor (IV) before amplification in a thermocycler (V). Transition between steps is performed by an operator.

**Figure 2 viruses-14-00313-f002:**
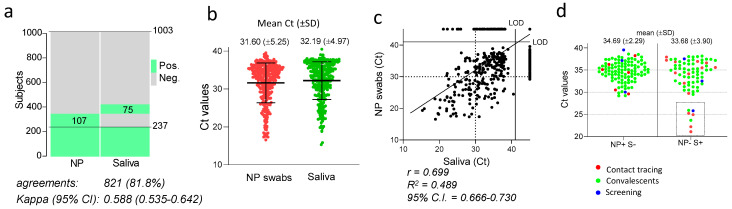
Global concordance between SARS-CoV-2 detection in saliva and NP swabs. (**a**) Heat map showing the individuals that tested positive (green) or negative (grey) for SARS-CoV-2 in the saliva or NP swabs. The percentage of agreements between results from saliva samples and NP swabs and the kappa index with 95% confidence interval (CI) is indicated. (**b**) Ct values of all positive NP swabs and saliva samples. Mean and standard deviation in indicated for each group. (**c**) Correlation between RT-PCR cycle threshold (Ct) values of paired nasopharyngeal swabs and self-administered saliva samples (*n* = 1003). Shown is the fitted curve of the linear regression, the Pearson correlation coefficient (r), the goodness of fit (R2) and the 95% confidence interval (CI). LOD: limit of detection. (**d**) Ct values of saliva samples and NP swabs from individuals testing positive with only one of the two methods, categorized by group of subjects as indicated by the colours specified in the legend. Indicated are the mean values and the standard deviations of the populations plotted.

**Figure 3 viruses-14-00313-f003:**
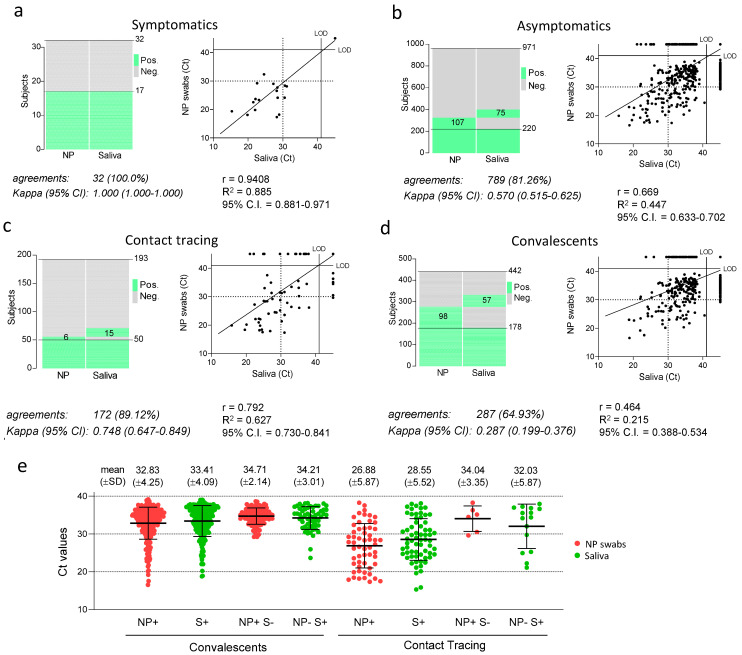
Concordance between SARS-CoV-2 detection in saliva samples and NP swabs in different groups of individuals tested. (**a**–**d**) In each panel, the heat maps show of individuals that tested positive and negative, and dot plots the RT-PCR cycle threshold (Ct) values of the paired nasopharyngeal swabs and self-administered saliva samples. The percentage of agreements between results from saliva samples and NP swabs and the kappa index with 95% confidence interval (CI) is indicated below the histograms, as well as the fitted curve of the linear regression in the dot plots, with the Pearson correlation coefficient (r), the goodness of fit (R^2^) and the 95% confidence interval (CI). LOD: limit of detection. (**e**) Ct values observed in NP swabs and saliva samples form total convalescent and contact tracing individuals (N+ and S+) as well as in discordant samples (NP+ S- and NP- S+). Indicated are mean values and standard deviations.

**Table 1 viruses-14-00313-t001:** Characteristics of the subjects recruited in the studies.

	n. of Individuals	Positive NP Swabs		Positive Saliva Samples	
		n	% (95% CI)	n	% (95% CI)
**Total**	1003	*344*	*34.3 (31.4–37.3)*	*312*	*31.1 (28.3–34.0)*
**Males**	434	*175*	*40 (35.8–45)*	*158*	*36.4 (32.0–41.0)*
**Females**	569	*169*	*29.7 (26.1–33.6)*	*154*	*27.1 (23.6–30.9)*
**Contact tracing**	193	*56*	*29.0 (22.7–36.0)*	*65*	*33.7 (27.4–40.6)*
**Convalescents**	442	*276*	*62.4 (57.8–66.8)*	*235*	*53.2% (48.5–57.8)*
**Screening**	351	*4*	*1.1 (0.3–3.0)*	*4*	*1.1 (0.3–3.0)*
**Suspected**	17	*8*	*47.1 (26.2–69.0)*	*8*	*47.1 (26.2–69.0)*

Age (years) mean +/− SD: 41.7 (+/− 15.2).

## Data Availability

All data generated and analyzed in this study are included in this Article. Additional data are available upon request from the corresponding author.

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
