# Peer review of "A Community Study of SARS-CoV-2 Detection by RT-PCR in Saliva: A Reliable and Effective Method"

_viruses, 2022, doi:10.3390/v14020313_

Round 1

Reviewer 1 Report

This is an interesting study and a well-written manuscript. Evaluating methods to streamline or simplify SARS-CoV-2 testing has clear public health benefits and implications. As we move into the next phase of the pandemic, and particularly in more resource limited settings or where health care resources are stretched, anything that can support efficient, cheap and easier SARS-CoV-2 testing or diagnosis will be a useful advance. There has also been much debate about the reliability of saliva testing for SARS-CoV-2, so a well executed and analyzed study such as this makes a  valuable contribution.  I have some relatively minor edits and suggestions below that would improve the manusscript.

1 The Introduction gives the impression that NP swabs are only performed by trained professionals and this is a limitation. Where I am located this is not the case and home PCR testing kits are available where people do their own at home NP swab for PCR. The introduction should be modified to address this. Furthermore, PCR testing also often involves a throat swab  but the manuscript gives the impression only the NP is swabbed. The manuscript text should be edited to reflect this where appropriate in the introduction/discussion.

2 Please improve Fig. 2C. it is not clear what the different groups are. Perhaps it is just the spacing of the axis labels, I assume data on left are saliva +ve NP-ve and on the right they are NP +ve and saliva -ve, maybe reducing the gap between the vertical labels would help.

3 Line 190 is truncated mid-sentence and text appears to be missing so the results section is incomplete??

4 Figure 3 text Line 213-214 (and also discussion) -  assumption is that it is the early stages of infection that makes the saliva test as reliable as NP swab, but the conclusion is based on the comparison convalescence in the text, but the asymptomatics were less concordant. It is equally likely the saliva results being concordant with NP is therefore related to viral load. I appreciate the authors may mean this too in lines 213-214 (with the assumption that early stages of infection naturally means higher viral load) but it is not explicitly stated. This text should be expanded to address this as a factor as it is postulated later in the discussion. Furthermore, what are the PCR Ct values for 3b (asymptomatic) who came up as negative for saliva but positive by NP swab? It would be interesting to see those data graphed separately to get a sense of whether lower viral loads results in false negative for those individual’s saliva tests. The authors comment on this idea of viral load in the discussion (lines 261-269) and Ct values are discussed, but the data are not easily visible and should be included in Fig. 3.

5 The discussion could expand on where/how the authors see the added value or implementation of this test would be useful? For example, how may it be rolled out? If people are going to a testing center then I don’t see saliva replacing the  NP swab by a professional, but do they propose this is a useful home test or is applicable to those harder to NP swab, such as children?

6 Is anything known about what SARS-CoV-2 strains the individuals were infected with? Is it likely that different VOCs with different properties (more transmissible, better replication) might perform differently? I expect these data are not available but this could be discussed at a minimum in the discussion when considering the advantages and disadvantages of the saliva test.

Minor edits

Line 82 replace difference with different

Line 85 errant remove . between screening (Table 1).

Line 89 sample collection not samples collection

Line 106, 108, 114 replace manufacturer with manufacturer’s

Table 1 what is meant by suspects? Clarify or correct wording to suspected?

Author Response

1 The Introduction gives the impression that NP swabs are only performed by trained professionals and this is a limitation. Where I am located this is not the case and home PCR testing kits are available where people do their own at home NP swab for PCR. The introduction should be modified to address this. Furthermore, PCR testing also often involves a throat swab  but the manuscript gives the impression only the NP is swabbed. The manuscript text should be edited to reflect this where appropriate in the introduction/discussion.

Thanks for suggesting this clarification. The text in abstract and introduction has now been modified to accommodate these useful observations, see lines 20-21, 45, 50-54. 326-327.

2 Please improve Fig. 2C. it is not clear what the different groups are. Perhaps it is just the spacing of the axis labels, I assume data on left are saliva +ve NP-ve and on the right they are NP +ve and saliva -ve, maybe reducing the gap between the vertical labels would help.

The graph has been modified accordingly, thanks for this suggestion. In addition, as part of the response to point 4, the average Ct values are now reported in the figure.

3 Line 190 is truncated mid-sentence and text appears to be missing so the results section is incomplete??

Apologies, an issue with the word processor resulted with the text being hidden below the figure. It has now been fixed.

4 Figure 3 text Line 213-214 (and also discussion) -  assumption is that it is the early stages of infection that makes the saliva test as reliable as NP swab, but the conclusion is based on the comparison convalescence in the text, but the asymptomatics were less concordant. It is equally likely the saliva results being concordant with NP is therefore related to viral load. I appreciate the authors may mean this too in lines 213-214 (with the assumption that early stages of infection naturally means higher viral load) but it is not explicitly stated. This text should be expanded to address this as a factor as it is postulated later in the discussion. Furthermore, what are the PCR Ct values for 3b (asymptomatic) who came up as negative for saliva but positive by NP swab? It would be interesting to see those data graphed separately to get a sense of whether lower viral loads results in false negative for those individual’s saliva tests. The authors comment on this idea of viral load in the discussion (lines 261-269) and Ct values are discussed, but the data are not easily visible and should be included in Fig. 3.

We agree with the reviewer, discordance is indeed associated with higher Ct values, as we now describe in the text and show in additional figures. The possibility that the discrepancy is linked to low viral loads, particularly in the convalescent group, has been discussed in the new version of the manuscript. For this reason, as suggested, figure 2b and figure 3e were added to compare Ct values in saliva and NP swabs globally and in the different groups of individuals. These figures are described in lines 200-201, and 241-249. The possibility of discordance being dependent on viral load is now described in discussion (lines 304-319).

5 The discussion could expand on where/how the authors see the added value or implementation of this test would be useful? For example, how may it be rolled out? If people are going to a testing center then I don’t see saliva replacing the  NP swab by a professional, but do they propose this is a useful home test or is applicable to those harder to NP swab, such as children?

Thanks for this suggestion which is now discussed in the revised manuscript (270-281)

6 Is anything known about what SARS-CoV-2 strains the individuals were infected with? Is it likely that different VOCs with different properties (more transmissible, better replication) might perform differently? I expect these data are not available but this could be discussed at a minimum in the discussion when considering the advantages and disadvantages of the saliva test.

We now specify in materials and methods that at the time the study was performed, the alpha variant was predominant (lines 126.127). The PCR kit used was known to perform efficiently for this variant. Lines 341-346 now comment on the use of the saliva test with the most recent and globally predominant variant.

Minor edits

Line 82 replace difference with different

Line 85 errant remove . between screening (Table 1).

Line 89 sample collection not samples collection

Line 106, 108, 114 replace manufacturer with manufacturer’s

Table 1 what is meant by suspects? Clarify or correct wording to suspected?

Many thanks for observing these errors which have now been corrected

Reviewer 2 Report

The authors developed an interesting saliva collection device and downstream automatic sample processing and RNA extraction followed by RT-PCR. Further, they analyzed a significant number of individuals (n=1003) and compared efficiency of SARS-CoV-2 detection with either the self-collected saliva or paired nasopharyngeal (NP) swabs, which are the gold standard diagnostic specimen for COVID-19. An overall moderate (68%) concordance was achieved between both specimens, where SARS-CoV-2 could be detected all symptomatic cases. Among asymptomatic individuals, saliva performed as well as NP swabs in individuals tested for contact tracing, but worse than NP swabs in convalescent individuals recovering from infection. Based on these results, authors argue that saliva is as sensitive as NP swabs to diagnose COVID-19 in early stages of infection. This is a straightforward work that further shows that saliva can be reliably used to detect infected individuals, particularly in contact tracing.

The manuscript is well written and results are clearly presented in general, and I enjoyed reading through the findings. However, I am not sure about the novelty of the work, since saliva has long been studied as an alternative diagnostic specimen to NP swabs. Could authors clarify here and within the manuscript what are the main novelties?

Besides that major concern, please find below other points that need to be addressed: 

1) Abstract, Line 22: Results were obtained from 1003 paired specimens, and not 1025 (after excluding 22 during quality control). Please change number to 1003.

2) Methods, Table 1: Please add lines to divide groups (e.g. Males and Females separated from Contact Tracing, Screening, etc). That would aid in visualization and interpretation.

3) Specimen collection needs some clarification. Line 96 says: 'NP samples were acquired over different days...'. Does this mean not all saliva and NP specimens were collected simultaneously?

4) Line 117: RdRP RT-PCR assays were used for viral detection. Is this test affected by mutations present in the main circulating variants in Italy at the time? 

5) Results, Line 182: Consider replacing 'good correlation' with 'strong correlation' or something similar. Correlation coefficients >0.7 usually indicate high or strong correlation.

6) Text portions are missing (Line 190). What are the conclusions for this paragraph?

7) Line 213: The Ct values in saliva and NP swabs in these groups are shown only for correlation purposes. Are average/median Ct values different between these specimens, in each group? This information could be used to better argue that saliva is as efficient as NP swabs during the early stages of infection, when viral load tends to be higher. Accordingly, Ct values seem to be higher in saliva than in NP swabs in the convalescent individuals. Could that indicate that viral clearance occurs earlier in saliva (or some other mechanism), leading to lower detection rates compared to NP swabs? 

This information could be added to enrich the results and discussion.

8) Discussion, Line 244 reads: 'This result suggests an inherent property of the biological samples that makes saliva an efficient matrix for the detection of the virus during the early stages of the infection and in
the presence of symptoms'. This statement sounds vague. Which inherent property could explain these results? Could salivary viral load be the main determinant of sensitivity here?

Author Response

The manuscript is well written and results are clearly presented in general, and I enjoyed reading through the findings. However, I am not sure about the novelty of the work, since saliva has long been studied as an alternative diagnostic specimen to NP swabs. Could authors clarify here and within the manuscript what are the main novelties?

We are aware that numerous reports have been already published on this topic, though sometimes presenting conflicting interpretations of the data. We think that the novelty of our study is twofold: the collection device and the design of the study itself. The intent of the study was first to test the performance of the novel device which facilitates both self-collection of saliva and downstream analysis. We have experienced that improving these combined factors impact enormously the analytical capacity and we think it is important to share this experience with the scientific community. On the other hand, the research was designed to test the performance of our strategy in a real-life context with individuals seeking diagnosis for different motivations, not necessarily in a clinical context. This we think is also a distinctive feature of our study.

These concepts are described in discussion (line 260-283)

Besides that major concern, please find below other points that need to be addressed: 

  • Abstract, Line 22: Results were obtained from 1003 paired specimens, and not 1025 (after excluding 22 during quality control). Please change number to 1003.

Many thanks, this was corrected in the revised version

2) Methods, Table 1: Please add lines to divide groups (e.g. Males and Females separated from Contact Tracing, Screening, etc). That would aid in visualization and interpretation.

Many thanks, this was implemented in the revised version

3) Specimen collection needs some clarification. Line 96 says: 'NP samples were acquired over different days...'. Does this mean not all saliva and NP specimens were collected simultaneously?

This sentence was not clear. It is now modified with: “More than one healthcare professional acquired the NP samples from different subjects”.

4) Line 117: RdRP RT-PCR assays were used for viral detection. Is this test affected by mutations present in the main circulating variants in Italy at the time? 

The vastly prevalent variant circulating at the time the research was conducted was alpha, which is well-detected by the PCR kit used (this is now stated in materials and methods, lines 126-127)

5) Results, Line 182: Consider replacing 'good correlation' with 'strong correlation' or something similar. Correlation coefficients >0.7 usually indicate high or strong correlation.

The modification was implemented, thanks for the suggestion

6) Text portions are missing (Line 190). What are the conclusions for this paragraph?

Apologies, an issue with the word processor resulted with the text being hidden below the figure. It has now been fixed

7) Line 213: The Ct values in saliva and NP swabs in these groups are shown only for correlation purposes. Are average/median Ct values different between these specimens, in each group? This information could be used to better argue that saliva is as efficient as NP swabs during the early stages of infection, when viral load tends to be higher. Accordingly, Ct values seem to be higher in saliva than in NP swabs in the convalescent individuals. Could that indicate that viral clearance occurs earlier in saliva (or some other mechanism), leading to lower detection rates compared to NP swabs? 

This information could be added to enrich the results and discussion.

This observation (a good point) was also raised by reviewer 1. Discordance is indeed associated with higher Ct values, as we now describe in the text and show in additional figures. The possibility that the discrepancy is linked to low viral loads, particularly in the convalescent group, has been discussed in the new version of the manuscript. For this reason, figure 2b and figure 3e were added to compare Ct values in saliva and NP swabs globally and in the different groups of individuals. These figures are described in lines 200-201, and 241-249. The possibility of discordance being dependent on viral load is now described in discussion (lines 304-319).

8) Discussion, Line 244 reads: 'This result suggests an inherent property of the biological samples that makes saliva an efficient matrix for the detection of the virus during the early stages of the infection and in
the presence of symptoms'. This statement sounds vague. Which inherent property could explain these results? Could salivary viral load be the main determinant of sensitivity here?

This point is related to the previous point and therefore the sentence has been removed as the discussion was expanded to better describe the possible implication of viral load in sensitivity of detection in saliva (lines 304-319).

Round 2

Reviewer 2 Report

The authors have  answered all my questions, and significant improvements have been made to the manuscript. I believe it can be accepted now, though very minor text revisions should be made (e.g. line 286 - DTT is a reducing agent; not 'anti-reducing'; Figure legends - 'heath map' should be heat map?).